# The Association of Vitamin D and Its Pathway Genes’ Polymorphisms with Hypertensive Disorders of Pregnancy: A Prospective Cohort Study

**DOI:** 10.3390/nu14112355

**Published:** 2022-06-06

**Authors:** Shuting Si, Minjia Mo, Haoyue Cheng, Zhicheng Peng, Xialidan Alifu, Haibo Zhou, Peihan Chi, Yan Zhuang, Yunxian Yu

**Affiliations:** 1Department of Public Health, and Department of Anesthesiology, Second Affiliated Hospital of Zhejiang University School of Medicine, Hangzhou 310058, China; 21818499@zju.edu.cn (S.S.); minjiamo@zju.edu.cn (M.M.); 3150101365@zju.edu.cn (H.C.); 22018678@zju.edu.cn (Z.P.); 3130100017@zju.edu.cn (X.A.); 11918158@zju.edu.cn (H.Z.); 22118872@zju.edu.cn (P.C.); yanzhuang@zju.edu.cn (Y.Z.); 2Department of Epidemiology & Health Statistics, School of Public Health, School of Medicine, Zhejiang University, Hangzhou 310058, China

**Keywords:** vitamin D, SNPs, hypertensive disorders of pregnancy, prospective cohort study

## Abstract

**Objective:** We aimed to explore the effect of single nucleotide polymorphism (SNP) in the genes of the vitamin D (VitD) metabolic pathway and its interaction with VitD level during pregnancy on the development of hypertensive disorders of pregnancy (HDP). **Methods:** The study was conducted in the Zhoushan Maternal and Child Health Care Hospital, China, from August 2011 to May 2018. The SNPs in VitD metabolic pathway-related genes were genotyped. Plasma 25-hydroxyvitamin vitamin D (25(OH)D) levels was measured at first (T1), second (T2), and third (T3) trimesters. The information of systolic blood pressure (SBP) and diastolic blood pressure (DBP), and the diagnosis of HDP were extracted from the electronic medical record system. Multivariable linear and logistic regression models and crossover analysis were applied. **Results:** The prospective cohort study included 3699 pregnant women, of which 105 (2.85%) were diagnosed with HDP. After adjusting for potential confounders, VitD deficiency at T2, as well as the change of 25(OH)D level between T1 and T2, were negatively associated with DBP at T2 and T3, but not HDP. Polymorphisms in *CYP24A1*, *GC*, and *LRP2* genes were associated with blood pressure and HDP. In addition, VitD interacted with *CYP24A1*, *GC*, and *VDR* genes’ polymorphisms on blood pressure. Furthermore, participants with polymorphisms in *CYP24A1*-rs2248137, *LRP2*-rs2389557, and *LRP2*-rs4667591 and who had VitD deficiency at T2 showed an increased risk of HDP. **Conclusions:** The individual and interactive association between VitD deficiency during pregnancy and SNPs in the genes of the VitD metabolic pathway on blood pressure and HDP were identified.

## 1. Introduction

Hypertensive disorders of pregnancy (HDP), including gestational hypertension, preeclampsia, eclampsia, pregnancy complicated with chronic hypertension, and chronic hypertension complicated with preeclampsia [1], accounted for nearly 18% of all maternal deaths worldwide [2]. Its increasing prevalence and related risks for maternal and child health as well as cardiovascular diseases later in life has garnered great attention in the field of public health [3,4]. The risk factors for HDP are advanced age, primipara, multiple pregnancy, family history of hypertension, high pre-pregnancy body mass index (BMI), and high basal blood pressure [5].

Approximately 5% to 7% of pregnancies are complicated by preeclampsia [6]. While the cause of preeclampsia is not fully discerned, previous studies have suggested that abnormal placentation and angiogenesis were central to the pathogenesis of this syndrome [6]. In recent years, growing evidence of the association between maternal hypovitaminosis D and increased risk of HDP has been suggested [7,8]. Compared to non-pregnant state, there are significant changes in vitamin D (VitD) metabolism during pregnancy, and the serum levels of VitD binding protein (VDBP) [9], as well as the active form, 1,25-dihydroxyvitamin (1,25(OH)_2_D) [10], increased notably. It is believed that not only the kidneys but also the placenta and decidua produce and secret 1,25(OH)_2_D during pregnancy [11]. Moreover, VitD receptors and related metabolic enzymes have been discovered in the placenta and decidua [12], indicating a potential role for VitD in implantation and placental function, outside of its well-established role in skeletal health [13]. 

To date, trial evidence appears insufficient to lean towards a protective effect of VitD supplementation during pregnancy against the risk of preeclampsia owing to small sample size or low study quality [14,15]. In addition, findings from observational studies in regard to the association between maternal VitD status and HDP are discrepant due to the large heterogeneity between study designs, lack of adherence to standardized outcome definitions, and different gestational weeks of VitD detection [8,16]. On the other hand, genetic variants in the VitD metabolic pathway have also been shown to participate in the pathogenesis of blood pressure increase and preeclampsia [8,17], which suggests a possible interaction between VitD and its pathway gene variants for HDP. The concentration or effect of VitD can be highly regulated due to the variation of key protein expression or activity. 25(OH)D is the main circulating metabolism and is considered the biological marker of VitD status. The main metabolic enzymes involved in the synthesis, transport, reabsorption, and inactivation of VitD include 25-hydroxylase (*CYP3A4*), 1-hydroxylase (*CYP27B1*), vitamin D-binding protein (*GC*), 24-hydroxylas and metaling (*LRP2*), and 24-hydroxylase (*CYP24A1*). Moreover, VitD receptor (*VDR*) regulates VitD metabolism through binding 1,25(OH)_2_D [18]. 

So far, most studies have only focused on the relationship between VitD status during pregnancy or gene variation in the VitD metabolic pathway and HDP, without considering the possible interaction between them. This study aimed to explore the association of VitD status in three trimesters of pregnancy with the risk of HDP, and to explore the interactive effect between maternal VitD level and genetic variants in the VitD metabolic pathways (*GC*, *CYP24A1*, *CYP3A4*, *CYP27B1*, *LRP2*, *VDR*) on gestational blood pressure and HDP.

## 2. Materials and Methods 

### 2.1. Study Design and Participants

The Zhoushan Pregnant Women Cohort (ZPWC) is an ongoing prospective cohort, conducted in Zhoushan Maternal and Child Health Care Hospital, Zhejiang. This study was based on the data of ZPWC from August 2011 to May 2018. We recruited pregnant women aged between 18 and 45 years at their first prenatal visit. A more detailed description of the inclusion and exclusion criteria can be seen in a previous study [19]. In addition, pregnant women without extreme/missing information of blood pressure and who measured plasma 25(OH)D levels in the first, second, or third trimester were included in the study. In addition, because gestational hypertension (GH), preeclampsia, and eclampsia are different from pregnancy complicated with chronic hypertension and chronic hypertension complicated with preeclampsia in pathogenesis and clinical treatment, pregnant women with chronic hypertension before pregnancy were also excluded [1]. Informed consent was obtained from all participants before the investigation. The study was conducted according to the guidelines of the Declaration of Helsinki and approved by the Institutional Review Board of Zhoushan Maternal and Child Health Care Hospital on 9 January 2011 (Ethical Approval Code: 2011-05).

### 2.2. Collection of Data and Blood Sample

The interviewers conducted face-to-face questionnaire surveys. Socio-demographic characteristics, lifestyle, and health behavior in the first (T1: 8th–14th gestational week), second (T2: 24th–28th gestational week), and third (T3: 32nd–36th gestational week) trimester, and 42nd day postpartum were collected. At each visit, professional nurses and inspectors were responsible for drawing and centrifuging fasting venous blood samples under 4 °C and separating the plasma and white blood cells, which were then stored at −80 °C until use. 

### 2.3. Measurement of 25(OH)D Concentrations

Plasma 25(OH)D_2_ and 25(OH)D_3_ concentrations (reported in ng/mL) were measured by Liquid chromatography–tandem mass spectrometry (API 3200MD (Applied Bio-systems/MDS Sciex, Framingham, MA, USA)). The lowest sensitivity of 25(OH)D_2_ and 25(OH)D_3_ was 2 ng/mL and 5 ng/mL, respectively. The intra-assay and inter-assay coefficient variance were 1.47–7.24% and 4.48–6.74% for 25(OH)D_2_ and 2.50–7.59% and 4.44–6.76% for 25(OH)D_3_, respectively [19]. The 25(OH)D concentrations were the sum of 25(OH)D_2_ and 25(OH)D_3_. 

### 2.4. Data Extraction

According to the guidelines of pregnant women prenatal health care, the first check-up and registration was conducted on the 8th–12th gestational week. After registration, 12 check-ups at 16, 20, 24, 28, 30, 32, 34, 36, 37, 38, 39, and 40 weeks of pregnancy were followed, along with a birth check every three days until delivery was performed after the 40th week, and a postpartum visit on the 42th day after delivery. The information including height, gestational age, and follow-up information (e.g., weight, systolic blood pressure (SBP), diastolic blood pressure (DBP), etc.), socio-demographic characteristics (e.g., age, education level, etc.), reproductive history (e.g., gravidity, parity, threatened abortion, and fetal malformation, etc.), history of present diseases (e.g., diabetes, etc.), pregnant complications (such as gestational diabetes mellitus, preeclampsia, and kidney disease, etc.), intrapartum complications (e.g., fetal distress, placenta previa, and placental abruption, etc.), was extracted from an electronic medical recorder system (EMRS). 

### 2.5. Covariates Assessment

According to Endocrine Society Clinical Practice Guidelines, we defined plasma 25(OH)D < 20 ng/mL (50 nmol/L) as VitD deficiency [20], and 25(OH)D concentrations ≥ 20 ng/mL as VitD non-deficiency. The change of 25(OH)D level during pregnancy is defined as a difference of 25(OH)D level between three trimesters. The following parameters were also defined: Pre-pregnancy body mass index (BMI) = weight (kg)/height^2^ (m^2^), gestational weight gain (continuous) = the weight on the day of VitD test at T1, T2, or T3, the pre-pregnancy weight, educational level (senior high school and below, college and above), gravity (1, ≥2, missing), parity (0, ≥1, missing), basal blood pressure (the level of blood pressure at the first prenatal examination or early pregnancy, continuous), the seasons of blood pressure measurement (divided as followed: spring (March to May), summer (June to August), fall (September to November), and winter (December to February) based on the sunshine intensity and duration in different months [21]).

### 2.6. HDP Definition

In perinatal care, SBP and DBP would be routinely measured [22]; we extracted the data from EMRS. In a sitting position, blood pressure measurement was performed from the right hand with a standard mercury sphygmomanometer. GH onset was defined as SBP ≥ 140 and/or DBP ≥ 90 mm Hg after the 20th gestational week (according to last menstruation date and B-ultrasound) in at least two consecutive examinations [23]. On the basis of GH, urinary protein ≥ +1 on a dipstick was defined as preeclampsia [1]. Eclampsia was defined as the presence of new-onset grand mal seizures in a woman with preeclampsia [24]. GH, preeclampsia, and eclampsia were combined as the group of HDP in later analysis.

### 2.7. SNP Selection and Genotyping

VitD-related SNP were selected if they met any one of the following conditions [25,26]: (1) SNPs positively associated with 25(OH)D concentration reported in the literature, and the minimum allele frequency (MAF) ≥ 10%; (2) SNPs displayed in the functional region in the NCBI database: exon region, intron splicing point, 5′end and 3′end regulatory regions, and MAF ≥ 10%; (3) HapMap Chinese database, including gene regions, SNPs within 1500 bp at the 5′end and 3′end; (4) selected by HaploView, the conditions are: MAF ≥ 10%; R^2^ ≥ 0.8. Finally, a total of 34 SNPs in the VitD metabolic pathway were selected (*CYP27B1*: rs10877012, *CYP3A4*: rs2242480, rs4646437, *LRP2*: rs4667591, rs10210408, rs2228171, rs7600336, rs2544381, rs2544390, rs2389557, *GC*: rs16846876, rs12512631, rs17467825, rs2070741, rs2282679, rs3755967, rs2298850, rs4588, rs7041, rs222020, rs1155563, rs2298849, *VDR*: rs2228570, rs7975232, rs11568820, rs2238136, rs2853559, rs4334089, rs10783219, *CYP24A1*: rs6013897, rs2762934, rs2209314, rs6127118, rs2248137).

The conventional phenol–chloroform extraction method was used to extract DNA from the peripheral blood leukocytes, which was then stored in TE-buffer at −80 °C. DNA was diluted to 10 ng/μL using a Nanodrop^®^ ND-1000 Spectrophotometer (Thermo Fisher Scientific Inc., Wilmington, NC, USA) for SNP analysis. A Sequenom MassARRAY iPLEX Gold platform (Sequenom, San Diego, CA, USA) was used for SNP genotyping. The call rate of these SNPs was over 98%, which conformed to the Hardy–Weinberg equilibrium.

### 2.8. Statistical Analysis

The characteristics between HDP and non-HDP groups were compared by t-test for continuous variables and by chi-squared test for categorical variables. Latent mixture modeling (PROC TRAJ) was used to identify subgroups that shared similar VitD patterns. Model fit was assessed using the Bayesian Information Criterion. We initiated a model with three trajectories, and then compared the BIC to that with two. The model with three trajectories identified fit best [27] (Appendix A). Restricted cubic spline (RCS) analyses were used to characterize the dose-response association and explore the potential linear or nonlinear relationship of 25(OH)D level in three trimesters, the change of 25(OH)D level during pregnancy with blood pressure in three trimesters, and HDP. Multivariable adjusted analyses with three knots were used. Test result for nonlinearity was checked first. If the test for nonlinearity was not significant, test result for overall association and linearity was checked, with a significant result indicating a linear association [28]. Multivariate adjusted RCS analysis showed that there was no nonlinear association of 25(OH)D level in three trimesters, the change of 25(OH)D level during pregnancy with blood pressure, and HDP during pregnancy (*P*_for non-linear_ > 0.05) (Appendix A). The Hardy–Weinberg equilibrium (HWE) of genotyped SNPs was tested using the χ^2^ test. 

A multiple linear regression model and a multivariate logistic regression model, combined with a crossover analysis method were utilized to explore the association between VitD and its metabolic pathway-related gene variants as well as their interactions with SBP, DBP, and HDP. The generalized linear model was used to analyze the relationship of the change of 25(OH)D level during pregnancy with SBP and DBP, and the multivariate logistic regression model was used to analyze the association between the changes in 25(OH)D levels and the trajectory of VitD during pregnancy with HDP. Models were adjusted for the following potential confounders: pre-pregnancy BMI, maternal age, gestational weight gain, gestational week, educational level, parity, basal blood pressure, and the seasons of blood pressure measurement. 

β (se) for linear regression, ORs, and corresponding 95% CIs for logistic regression were calculated, respectively. All test results were considered statistically significant at a value of *p* < 0.05. RCS analyses were performed using R software (version 3.6.3); the other analyses were performed using SAS (version 9.4, SAS Institute, Cary, NC, USA).

## 3. Results

### 3.1. Subject Characteristics

The demographic characteristics of participants with HDP or non-HDP were compared and are shown in Table 1. The prospective cohort study included 3699 pregnant women, of which 105 (2.85%) were diagnosed with HDP. The mean age was 29.30 ± 3.95 years for HDP participants and 28.67 ± 3.64 years for non-HDP participants. Compared with non-HDP participants, HDP women had higher pre-pregnancy BMI (21.16 ± 2.91 kg/m^2^ vs. 23.62 ± 4.05 kg/m^2^, *p* < 0.0001). The SBP and DBP levels in three trimesters were higher in HDP than non-HDP. However, VitD deficiency in three trimesters, educational level, gravity, and parity were not significantly different between the two groups. The characteristics of participants in the SNP analysis are shown in Table 1. Pregnant women with HDP had higher pre-pregnancy BMI than the non-HDP group. There was no significant difference in weight gain and 25(OH)D level in three trimesters, educational level, gravity, and parity between the two groups (Appendix A).

### 3.2. The Association between 25(OH)D in Three Trimesters and HDP

After being adjusted for potential confounders, 25(OH)D level at T1 was negatively associated with SBP (β (se) = −0.05 (0.02), *p* = 0.0287) and DBP (β (se) = −0.05 (0.02), *p* = 0.0190) at T1. In addition, 25(OH)D level at T2 was negatively associated with DBP at T2 and T3, respectively (β (se) = −0.10 (0.02), *p* < 0.0001, β (se) = −0.07 (0.02), *p* = 0.0003) (Table 2). The association between VitD deficiency in three trimesters with SBP and DBP were consistent with the above results (Appendix A). For each 1 ng/mL increase in 25(OH)D changes between T1 and T2, DBP at T2 and T3 decreased by 0.11 (se = 0.02) mmHg and 0.11 (se = 0.02) mmHg, respectively (*p* < 0.0001). (Table 3). Three subgroups of participants with data of 25(OH)D levels in three trimesters were identified by latent mixture modeling. Compared with women whose 25(OH)D levels remained low from T1 to T3, women whose 25(OH)D levels gradually increased at T2 and T3 or whose 25(OH)D levels remained high during pregnancy had lower DBP at T3 (β (se) = −1.13 (0.46), *p* = 0.0137, β (se) = −1.74 (0.74), *p* = 0.0195) (Table 4). However, there was no significant association between 25(OH)D levels, VitD deficiency in three trimesters, the change of 25(OH)D levels, or the VitD trajectory during pregnancy with HDP (Appendix A).

### 3.3. The Association between SNP and HDP

The association of each SNP genotype with SBP and DBP at T1, T2, and T3 are shown in Appendix A, respectively. Polymorphisms in *CYP24A1*-rs2248137 was significantly associated with higher SBP at T1 and DBP at T2 and T3. Polymorphisms in *CYP24A1*-rs2762934 were significantly associated with higher DBP at T1 and SBP at T2. Polymorphisms in *LRP2*-rs4667591 were significantly associated with higher SBP at T1 and DBP at T3. Polymorphisms in *GC*-rs2070741, rs222020, and rs2298849 were associated with higher SBP at T2. Polymorphisms in *LRP2*-rs2544390 were associated with higher DBP at T3. Furthermore, polymorphisms in *CYP24A1*-rs2248137, *CYP24A1*-rs2762934, *CYP24A1*-rs6127118, and *GC*-rs2070741 were associated a higher risk of HDP (Table 5). However, there was no significant association between other genes’ polymorphisms and HDP.

### 3.4. The Interaction between Single SNP and VitD Deficiency in Three Trimesters on the Risk of HDP

Results of the crossover analysis are shown in Appendix A. Polymorphisms of seven SNPs (rs16846876, rs2282679, rs17467825, rs2298849, rs2298850, rs3755967, and rs4588) in *GC* gene and VitD deficiency at T2 might exert interactions on DBP at T2. In addition, *VDR*-rs2228570 and VitD deficiency at T2 might exert interaction on SBP at T2. Furthermore, women with mutations in *CYP24A1*-rs2248137, *LRP2*-rs2389557, and *LRP2*-rs4667591 and had VitD deficiency at T2 showed increased risk of HDP (Table 6).

## 4. Discussion

In the present study, 25(OH)D level at T2, as well as 25(OH)D change between T1 and T2, were significantly inversely associated with DBP at T2 and T3. However, significant associations between maternal VitD deficiency in any trimesters and HDP were not observed. Polymorphism in *CYP24A1*, *GC*, and *LRP2* was associated with blood pressure, and polymorphism in *CYP24A1* and *GC* was associated with increased risk of HDP. Furthermore, interactive effects between VitD deficiency and polymorphisms in *CYP24A1*, *GC*, and *VDR* genes on blood pressure were identified. Women with polymorphisms in *CYP24A1* and *LRP2* genes and had VitD deficiency at T2 showed a higher risk of HDP.

Previous findings on the association between VitD level during pregnancy and HDP were not consistent. A prospective observational study conducted in southern China found that there were no significant differences in the risk of HDP among women with different levels of VitD at 16–20-week gestation [29]. A case-control study conducted in Iran found that pregnant women with VitD deficiency (25(OH)D < 20 ng/mL) had higher blood pressure and increased risk of preeclampsia than those with VitD insufficiency (25(OH)D: 20~30 ng/mL) [8]. The prospective Swedish GraviD cohort study, including 1413 pregnant women, found that 25(OH)D was positively associated with T1 blood pressure [16]; however, both 25(OH)D level at T3 and change in 25(OH)D level from T1 to T3 were significantly and negatively associated with preeclampsia, but not with the risk of GH [30]. Another nested case-control study carried out among Australian pregnant women found that higher levels of VitD (25(OH)D > 75 nmol/L) in early pregnancy (10–14 weeks) could prevent the occurrence of early-onset preeclampsia (*p* = 0.09); however, women with low levels of 25(OH)D (<37.5 ng/mL) in the first trimester of pregnancy had a tendency toward reduced risk of preeclampsia (*p* = 0.07) [31]. Conflicting data for an association of VitD during pregnancy with HDP results from a number of sources, including large heterogeneity between study designs, different ethnicities, different subtypes of HDP included in the analysis, variable quality of measurement for 25(OH)D, and inconsistent definition of VitD status [32]. On the other hand, studies have shown that the gene variation of key enzymes in VitD synthesis, transport and metabolism pathway would also affect the levels and effects of 25(OH)D and 1,25(OH)_2_D [25,33]. Furthermore, genetic mutations in the VitD metabolic pathway were also associated with increased risk of HDP [8].

The active form of VitD (1,25(OH)_2_D) needs to bind to VDR to exert its biological function. Relevant studies related to genetic variants in the VitD metabolic pathway with HDP were mainly focus on three SNPs (rs2228570, rs731236, and rs1544410) of *VDR* gene. Rezavand et al. [8] found that, compared with *VDR*-rs2228570 TC and TT + TC genotypes, the SBP and DBP of CC genotype were higher, and the risk of preeclampsia increased by 1.72 times. However, no association was found between *VDR*-rs731236, *VDR*-rs1544410, and preeclampsia. Knabl et al. [34] also reported that there was a strong association between the polymorphisms in rs10735810 and rs1544410 of *VDR* and the risk of GH. The polymorphisms in rs10735810 affect plasma renin activity and may be associated with a reduced risk of GH [34]. In this study, VitD deficiency at T2 interacted with the variants of *VDR*-rs2238136 on DBP and *VDR*-rs2228570 on SBP at T2.

The *CYP24A1* gene is located in 20q13-2, which is mainly expressed in the kidney and encodes the catabolic enzymes of 1,25(OH)_2_D and 25(OH)D [35]. Evidence relating to the association between *CYP24A1* gene polymorphism and susceptibility to hypertension, especially among pregnant women, is scare. A case-control study among the Chinese Han population found that *CYP24A1*-rs56229249 significantly decreased the hypertension risk in homozygote and recessive models [36]. In addition, rs2762940 was related to hypertension risk in men, and rs56229249 was a protective factor against hypertension in women [36]. The comprehensive genetic association study in the Women’s Genome Health Study (WGHS) found that *CYP24A1*-rs2296241 showed significant associations with SBP, DBP, mean arterial pressure, and pulse pressure [37]. In this study, we found that gene variants in *CYP24A1*-rs2248137, *CYP24A1*-rs2762934, and *CYP24A1*-rs6127118 were associated with increased risk of HDP. Furthermore, *CYP24A1*-rs6013897 interacted with VitD deficiency at T2 on HDP. On the other hand, *LRP2* is located on 2q24-q31, which is a member of the low-density lipoprotein receptor family and encodes megalin protein. In the kidney, megalin and cubilin combine together with hydroxylate 25(OH)D_3_ into 1,25(OH)_2_D_3_ [38]. Studies regarding the association between *LRP2* genes and VitD with the risk for HDP are still lacking. This study found that the mutations of *LRP2*-rs2389557 and *LRP2*-rs4667591 and VitD deficiency at T2 had a combined effect on the risk of HDP.

The *GC* gene encodes VitD binding protein (VDBP) [39], which is the major transporter of VitD. About 85% to 90% of 25(OH)D is bound to VDBP in circulation [40]. VDBP can aggravate or enhance various biological processes during pregnancy, such as immune regulation, glucose metabolism, and blood pressure regulation [39]. The *GC*-1 subtype was more common in pregnant women with preeclampsia than in those without preeclampsia, which was considered as a potential early detection genetic marker for women at risk of preeclampsia [41]. In HIV endemic areas of South Africa, compared with women with normal blood pressure, two SNPs of *GC* gene (rs4588 and rs7041) were more common in pregnant women with preeclampsia, and were not related to HIV status [42]. Furthermore, *GC*-rs4588 polymorphism was associated with early-onset (<34 weeks) and late-onset (≥34 weeks of pregnancy) preeclampsia, while *GC*-rs7041 was associated with early-onset eclampsia [42]. A nested case-control study of 170 American women from Massachusetts tracked the levels of VDBP and 25(OH)D throughout pregnancy to examine whether these biomarkers were associated with blood pressure or the risk of preeclampsia, but found no significant correlation of VDBP or 25(OH)D levels with preeclampsia [43]. At present, the combined effect of *GC* gene polymorphism and VitD during pregnancy on HDP is not clear. A study focused on preterm birth found that rs7041 variants interacted with VitD at T2 on the gestational week of delivery and preterm birth [44]. Our study found that the variant of *GC*-rs2070741 was associated with higher SBP at T2 and increased risk of HDP. Mutations at *GC* rs16846876, rs2282679, rs17467825, rs2298849, rs2298850, rs3755967, and rs4588 interacted with VitD deficiency at T2 on higher DBP at T2.

To our knowledge, this is the first prospective cohort study exploring the association between VitD in three trimesters and VitD pathway gene variants as well as their interactions on SBP, DBP, and the risk of HDP. However, limitations could not be neglected. First of all, some subjects had a lack of 25(OH)D data at T2 and T3, and therefore selection bias might exist. However, subgroup analysis of pregnant women with VitD detected at T1 and T2 showed that the results were almost consistent with the results in the whole study population. Secondly, as the prevalence of HDP in this study was relatively low (2.84%), the association between VitD and different HDP subtypes (GH, preeclampsia, eclampsia) could not be explored. However, studies have shown that, although these subtypes can appear alone, they are progressive manifestations of a single process and share common etiology [45,46]. Lastly, the relatively single ethnic population of this study may also limit the extrapolation of findings.

## 5. Conclusions

This study found that the level of 25(OH)D at T1 and T2 was negatively correlated with DBP at T2. In addition, polymorphisms in VitD metabolic pathway genes, including *CYP24A1* and *GC*, increased the risk of HDP. Furthermore, gene variants in *CYP24A1* and *LRP2* and VitD deficiency at T2 showed combined effect on the risk of HDP, but the specific mechanism remains to be further investigated. The results of this study provide a scientific basis for the clinical detection of VitD during pregnancy and the supplementation of VitD during pregnancy.

## Figures and Tables

**Table 1 nutrients-14-02355-t001:** Baseline characteristics of pregnant women.

Variables	Non-HDP (N = 3594)	HDP (N = 105)	*p*
Mean ± SD			
**Age, years**	28.67 ± 3.64	29.30 ± 3.95	0.0811
**Pre-pregnancy BMI, kg/m^2^**	21.16 ± 2.91	23.62 ± 4.05	<0.0001
**T1 (N = 3302)**			
Weight gain, kg	0.01 ± 0.17	0.02 ± 0.14	0.3841
SBP, mmHg	103.53 ± 9.31	112.21 ± 10.58	<0.0001
DBP, mmHg	68.35 ± 6.67	74.48 ± 5.99	<0.0001
25(OH)D, ng/mL	17.85 ± 8.38	17.70 ± 7.10	0.8629
**T2 (N = 2479)**			
Weight gain, kg	5.61 ± 3.82	6.25 ± 4.50	0.1971
SBP, mmHg	107.17 ± 9.24	116.60 ± 14.44	<0.0001
DBP, mmHg	69.13 ± 7.82	76.61 ± 8.84	<0.0001
25(OH)D, ng/mL	23.28 ± 10.38	22.91 ± 9.60	0.7827
**T3 (N = 1549)**			
Weight gain, kg	11.91 ± 3.73	11.20 ± 4.69	0.2181
SBP, mmHg	108.86 ± 9.72	123.30 ± 15.68	<0.0001
DBP, mmHg	70.99 ± 7.38	82.43 ± 8.44	<0.0001
25(OH)D, ng/mL	26.53 ± 11.28	26.20 ± 11.01	0.8468
N (%)			
**VitD deficiency at T1 ^a^**	2176 (67.85)	66 (69.47)	0.7386
**VitD deficiency at T2 ^b^**	1067 (44.15)	30 (48.39)	0.5067
**VitD deficiency at T3 ^c^**	476 (31.63)	13 (29.55)	0.7696
**Educational level**			0.1263
≤High school	957 (26.63)	35 (33.33)	
>High school	2637 (73.37)	70 (66.67)	
**Gravity**			0.5389
1	1652 (45.97)	43 (40.95)	
≥2	1822 (50.70)	59 (56.19)	
Unknown	120 (3.34)	3 (2.86)	
**Parity**			0.4887
0	2015 (56.07)	65 (61.90)	
≥1	771 (21.45)	20 (19.05)	
Unknown	808 (22.48)	20 (19.05)	

Abbreviations: HDP, hypertensive disorders in pregnancy; BMI, body mass index; SBP, systolic blood pressure; DBP, diastolic blood pressure; VitD, vitamin D. ^a^ N = 3302, ^b^ N = 2479, ^c^ N = 1549.

**Table 2 nutrients-14-02355-t002:** Association between 25(OH)D levels in three trimesters with blood pressure.

Trimesters of 25(OH)D	N	SBP, mmHg		DBP, mmHg	
		β (se)	*p*	β (se)	*p*
		**Blood pressure at T1 (N = 3302)**			
T1	3302	−0.02 (0.02)	0.3198	0.02 (0.01)	0.2056
		**Blood pressure at T2 (N = 2479)**			
T1	2125	−0.05 (0.02)	0.0287	−0.05 (0.02)	0.0190
T2	2479	0.03 (0.02)	0.1675	−0.10 (0.02)	<0.0001
		**Blood pressure at T3 (N = 1549)**			
T1	1328	0.03 (0.03)	0.2361	−0.02 (0.02)	0.3259
T2	1390	0.04 (0.03)	0.1214	−0.07 (0.02)	0.0003
T3	1549	0.04 (0.02)	0.0541	−0.02 (0.02)	0.2486

Abbreviations: SBP, systolic blood pressure, DBP, diastolic blood pressure. Adjusted for pre-pregnancy BMI, maternal age, gestational weight gain, gestational week, educational level, parity, basal blood pressure, and the seasons of blood pressure measurement.

**Table 3 nutrients-14-02355-t003:** The association between the change of 25(OH)D levels during pregnancy and blood pressure at T2 and T3.

The Change of Trimesters	N	The Change of 25(OH)D Levels, ng/mL *	SBP, mmHg		DBP, mmHg	
			β (se)	*p*	β (se)	*p*
			**Blood pressure at T2 (N = 2479)**			
Between T1 and T2	2125	3.50 (84.59)	0.03 (0.02)	0.1217	−0.11 (0.02)	<0.0001
			**Blood pressure at T3 (N = 1549)**			
Between T1 and T2	1212	2.40 (81.27)	0.03 (0.03)	0.3142	−0.11 (0.02)	<0.0001
Between T1 and T3	1328	6.59 (98.02)	0.06 (0.03)	0.0294	−0.02 (0.02)	0.3516
Between T2 and T3	1390	3.40 (87.23)	0.02 (0.03)	0.3662	0.04 (0.02)	0.0405

Abbreviations: SBP, systolic blood pressure; DBP, diastolic blood pressure. * Presented as the median (range). Adjusted for pre-pregnancy BMI, maternal age, gestational weight gain, gestational week, educational level, parity, basal blood pressure, the seasons of blood pressure measurement, and 25(OH)D level at T1.

**Table 4 nutrients-14-02355-t004:** The association between the trajectory of VitD during pregnancy and blood pressure at T3.

Trajectory of VitD	N (%)	SBP, mmHg		DBP, mmHg	
		β (se)	*p*	β (se)	*p*
Subgroup 1	621 (51.24)	Ref		Ref	
Subgroup 2	469 (38.70)	0.48 (0.60)	0.4216	−1.13 (0.46)	0.0137
Subgroup 3	122 (10.07)	1.58 (0.98)	0.1052	−1.74 (0.74)	0.0195

Abbreviations: SBP, systolic blood pressure; DBP, diastolic blood pressure; VitD, vitamin D. Adjusted for pre-pregnancy BMI, maternal age, gestational weight gain, gestational week, educational level, parity, basal blood pressure, and the seasons of blood pressure measurement.

**Table 5 nutrients-14-02355-t005:** The relationship between single SNP and HDP *.

SNP	Genotypes	N	Case (%)	Crude Model		Adjusted Model *	
				OR (95%CI)	*p*	OR (95%CI)	*p*
** *CYP24A1* **							
rs2209314	TT	941	28 (3.0)	Ref		Ref	
	CT	1309	38 (2.9)	0.97 (0.59–1.60)	0.9198	0.99 (0.60–1.64)	0.9648
	CC	443	4 (0.9)	0.30 (0.10–0.85)	0.024	0.30 (0.10–0.87)	0.026
rs2248137	GG	934	18 (1.9)	Ref		Ref	
	GC	453	20 (4.4)	2.35 (1.23–4.49)	0.0096	2.62 (1.32–5.21)	0.0059
	CC	643	19 (3.0)	1.55 (0.81–2.98)	0.1885	1.80 (0.92–3.53)	0.0869
rs2762934	GG	599	22 (3.7)	Ref		Ref	
	GA	119	5 (4.2)	1.15 (0.43–3.10)	0.7819	1.07 (0.38–3.00)	0.9051
	AA	7	1 (14.3)	4.37 (0.50–37.88)	0.1806	9.98 (1.06–94.04)	0.0444
rs6013897	TT	529	20 (3.8)	Ref		Ref	
	AT	172	8 (4.7)	1.24 (0.54–2.87)	0.6131	1.26 (0.53–3.02)	0.5964
	AA	22	1 (4.5)	1.21 (0.16–9.46)	0.8546	1.53 (0.19–12.57)	0.6905
rs6127118	GG	963	24 (2.5)	Ref		Ref	
	AG	1616	41 (2.5)	1.02 (0.61–1.70)	0.9439	0.96 (0.57–1.61)	0.8736
	AA	119	7 (5.9)	2.45 (1.03–5.80)	0.0426	2.38 (0.98–5.77)	0.0542
** *CYP27B1* **							
rs10877012	TT	1125	27 (2.4)	Ref		Ref	
	GT	1204	32 (2.7)	1.11 (0.66–1.87)	0.6925	1.14 (0.67–1.93)	0.6354
	GG	360	12 (3.3)	1.40 (0.70–2.80)	0.3366	1.61 (0.80–3.25)	0.1856
** *CYP3A4* **							
rs2242480	CC	1529	41 (2.7)	Ref		Ref	
	CT	1004	26 (2.6)	0.96 (0.59–1.59)	0.8879	0.96 (0.58–1.60)	0.8885
	TT	159	5 (3.1)	1.18 (0.46–3.03)	0.7329	1.37 (0.53–3.56)	0.5164
rs4646437	GG	530	22 (4.2)	Ref		Ref	
	AG	182	7 (3.8)	0.92 (0.39–2.20)	0.8576	0.83 (0.34–2.04)	0.6868
** *GC* **							
rs1155563	TT	951	29 (3.0)	Ref		Ref	
	TC	1290	31 (2.4)	0.78 (0.47–1.31)	0.3499	0.76 (0.45–1.29)	0.3091
	CC	450	11 (2.4)	0.80 (0.39–1.61)	0.5262	0.73 (0.36–1.50)	0.3886
rs12512631	TT	463	21 (4.5)	Ref		Ref	
	CT	241	7 (2.9)	0.63 (0.26–1.50)	0.2973	0.77 (0.31–1.89)	0.5683
	CC	18	1 (5.6)	1.24 (0.16–9.75)	0.8393	1.07 (0.13–8.84)	0.9466
rs16846876	AA	1274	38 (3.0)	Ref		Ref	
	AT	1133	25 (2.2)	0.73 (0.44–1.22)	0.2357	0.69 (0.41–1.16)	0.1632
	TT	292	9 (3.1)	1.03 (0.49–2.16)	0.9283	0.89 (0.41–1.93)	0.7745
rs17467825	AA	1254	37 (3.0)	Ref		Ref	
	GA	1150	29 (2.5)	0.85 (0.52–1.39)	0.5208	0.80 (0.48–1.32)	0.3792
	GG	299	6 (2.0)	0.67 (0.28–1.61)	0.375	0.54 (0.22–1.33)	0.181
rs2070741	TT	495	17 (3.4)	Ref		Ref	
	GT	213	8 (3.8)	1.10 (0.47–2.58)	0.8317	1.10 (0.46–2.66)	0.8251
	GG	18	3 (16.7)	5.62 (1.49–21.28)	0.011	4.77 (1.12–20.21)	0.0341
rs222020	TT	270	10 (3.7)	Ref		Ref	
	CT	344	12 (3.5)	0.94 (0.40–2.21)	0.8867	0.85 (0.35–2.07)	0.7284
	CC	110	6 (5.5)	1.50 (0.53–4.23)	0.4436	1.42 (0.49–4.15)	0.5169
rs2282679	TT	1254	36 (2.9)	Ref		Ref	
	GT	1141	30 (2.6)	0.91 (0.56–1.49)	0.7185	0.87 (0.52–1.43)	0.5726
	GG	306	6 (2.0)	0.68 (0.28–1.62)	0.3808	0.55 (0.22–1.35)	0.1913
rs2298849	AA	1120	27 (2.4)	Ref		Ref	
	GA	1219	38 (3.1)	1.30 (0.79–2.15)	0.3003	1.32 (0.79–2.20)	0.2835
	GG	366	7 (1.9)	0.79 (0.34–1.83)	0.5809	0.82 (0.35–1.90)	0.6376
rs2298850	GG	1229	35 (2.8)	Ref		Ref	
	CG	1151	29 (2.5)	0.88 (0.54–1.45)	0.621	0.84 (0.50–1.39)	0.4891
	CC	305	6 (2.0)	0.68 (0.29–1.64)	0.3967	0.55 (0.22–1.38)	0.2051
rs3755967	CC	1250	36 (2.9)	Ref		Ref	
	CT	1149	30 (2.6)	0.90 (0.55–1.48)	0.6875	0.86 (0.52–1.42)	0.5515
	TT	306	6 (2.0)	0.67 (0.28–1.62)	0.3768	0.54 (0.22–1.35)	0.189
rs4588	GG	1241	36 (2.9)	Ref		Ref	
	GT	1146	29 (2.5)	0.87 (0.53–1.43)	0.5789	0.83 (0.50–1.37)	0.4576
	TT	306	6 (2.0)	0.67 (0.28–1.60)	0.3679	0.54 (0.22–1.34)	0.1819
rs7041	AA	1445	39 (2.7)	Ref		Ref	
	CA	1061	28 (2.6)	0.98 (0.60–1.60)	0.9268	1.03 (0.62–1.70)	0.9026
	CC	195	4 (2.1)	0.76 (0.27–2.14)	0.5968	0.87 (0.30–2.48)	0.7934
** *LRP2* **							
rs10210408	CC	895	22 (2.5)	Ref		Ref	
	TC	1325	40 (3.0)	1.24 (0.73–2.09)	0.4322	1.16 (0.68–1.99)	0.5816
	TT	486	10 (2.1)	0.83 (0.39–1.78)	0.6378	0.81 (0.38–1.74)	0.5919
rs2228171	TT	932	24 (2.6)	Ref		Ref	
	CT	394	15 (3.8)	1.50 (0.78–2.89)	0.2279	1.48 (0.75–2.94)	0.2607
	CC	301	8 (2.7)	1.03 (0.46–2.32)	0.9375	1.14 (0.50–2.61)	0.7534
rs2389557	AA	194	8 (4.1)	Ref		Ref	
	GA	363	12 (3.3)	0.79 (0.32–1.98)	0.6218	0.78 (0.31–2.00)	0.6094
	GG	166	9 (5.4)	1.33 (0.50–3.54)	0.5639	1.24 (0.45–3.42)	0.6722
rs2544381	GG	388	20 (5.2)	Ref		Ref	
	CG	284	7 (2.5)	0.46 (0.19–1.12)	0.0862	0.48 (0.19–1.18)	0.108
	CC	52	1 (1.9)	0.36 (0.05–2.75)	0.3249	0.37 (0.05–2.92)	0.349
rs2544390	CC	199	9 (4.5)	Ref		Ref	
	CT	370	14 (3.8)	0.83 (0.35–1.95)	0.6698	0.70 (0.29–1.71)	0.4308
	TT	154	5 (3.2)	0.71 (0.23–2.16)	0.5442	0.68 (0.22–2.15)	0.5131
rs4667591	TT	245	10 (4.1)	Ref		Ref	
	GT	347	11 (3.2)	0.77 (0.32–1.84)	0.5558	0.82 (0.33–2.03)	0.671
	GG	132	8 (6.1)	1.52 (0.58–3.94)	0.3929	1.68 (0.62–4.56)	0.3081
rs7600336	CC	236	8 (3.4)	Ref		Ref	
	TC	342	14 (4.1)	1.22 (0.50–2.95)	0.6643	1.10 (0.44–2.75)	0.8428
	TT	148	6 (4.1)	1.20 (0.41–3.54)	0.7357	1.30 (0.43–3.93)	0.6374
** *VDR* **							
rs10783219	AA	998	26 (2.6)	Ref		Ref	
	TA	1275	36 (2.8)	1.09 (0.65–1.81)	0.7512	1.03 (0.61–1.74)	0.8982
	TT	428	9 (2.1)	0.80 (0.37–1.73)	0.5753	0.76 (0.35–1.66)	0.4869
rs11568820	CC	219	6 (2.7)	Ref		Ref	
	TC	350	17 (4.9)	1.81 (0.70–4.67)	0.2182	1.93 (0.72–5.14)	0.19
	TT	153	5 (3.3)	1.20 (0.36–4.00)	0.7675	1.48 (0.43–5.14)	0.5328
rs2228570	GG	212	10 (4.7)	Ref		Ref	
	GA	364	15 (4.1)	0.87 (0.38–1.97)	0.7351	0.86 (0.37–2.00)	0.7204
	AA	148	3 (2.0)	0.42 (0.11–1.55)	0.191	0.41 (0.11–1.56)	0.1905
rs2238136	CC	483	19 (3.9)	Ref		Ref	
	TC	218	10 (4.6)	1.17 (0.54–2.57)	0.6879	1.16 (0.51–2.60)	0.7259
rs2853559	GG	319	16 (5.0)	Ref		Ref	
	GA	313	10 (3.2)	0.62 (0.28–1.40)	0.2531	0.68 (0.29–1.56)	0.3599
	AA	87	2 (2.3)	0.45 (0.10–1.98)	0.2875	0.40 (0.09–1.84)	0.2418
rs4334089	GG	227	7 (3.1)	Ref		Ref	
	AG	350	15 (4.3)	1.41 (0.56–3.51)	0.4637	1.45 (0.56–3.75)	0.4397
	AA	148	6 (4.1)	1.33 (0.44–4.03)	0.617	1.54 (0.49–4.85)	0.4563
rs7975232	CC	382	11 (2.9)	Ref		Ref	
	CA	285	15 (5.3)	1.87 (0.85–4.14)	0.121	1.84 (0.81–4.20)	0.1458
	AA	60	3 (5.0)	1.78 (0.48–6.56)	0.3894	1.77 (0.46–6.78)	0.4031

Abbreviations: VitD, vitamin D; HDP, hypertensive disorders in pregnancy. * Adjusted for pre-pregnancy BMI, maternal age, gestational weight gain, educational level, parity, and basal blood pressure.

**Table 6 nutrients-14-02355-t006:** The interaction between SNPs and VitD in three trimesters on the risk of HDP.

SNP	Genotypes	VitD	T1		T2		T3	
			N	OR (95%CI)	N	OR (95%CI)	N	OR (95%CI)
** *CYP24A1* **								
rs2209314	CC/CT	≥20	545	Ref	724	Ref	557	Ref
	TT	≥20	316	1.60 (0.66–3.87)	383	1.41 (0.65–3.08)	294	1.86 (0.80–4.28)
	CC/CT	<20	1026	1.18 (0.57–2.45)	541	1.13 (0.53–2.40)	260	1.20 (0.43–3.29)
	TT	<20	560	1.52 (0.70–3.32)	309	1.41 (0.62–3.18)	145	1.04 (0.28–3.79)
	*P* _interaction_			0.8774		0.7996		0.5739
rs2248137	GG	≥20	357	Ref	451	Ref	339	Ref
	GC	≥20	117	2.50 (0.81–7.73)	111	4.45 (1.25–15.79) *	106	3.81 (1.11–13.07) *
	CC	≥20	220	1.04 (0.32–3.31)	269	3.90 (1.33–11.41) *	195	2.22 (0.66–7.47)
	GG	<20	480	0.77 (0.28–2.09)	283	2.11 (0.67–6.63)	131	1.17 (0.22–6.23)
	GC	<20	317	1.80 (0.70–4.67)	133	5.42 (1.71–17.23) *	63	1.05 (0.12–9.51)
	CC	<20	377	1.65 (0.64–4.26)	171	1.17 (0.22–6.04)	90	2.47 (0.56–10.81)
	*P* _interaction_			0.5195		0.1442		0.8045
rs6127118	GG	≥20	288	Ref	405	Ref	316	Ref
	AG/AA	≥20	572	2.01 (0.66–6.10)	702	0.83 (0.38–1.81)	537	2.97 (0.99–8.88)
	GG	<20	582	2.29 (0.77–6.87)	291	0.77 (0.28–2.12)	133	1.86 (0.40–8.58)
	AG/AA	<20	1010	1.63 (0.56–4.77)	558	1.12 (0.52–2.42)	271	2.18 (0.62–7.65)
	*P* _interaction_			0.855		0.2332		0.5262
** *CYP27B1* **								
rs10877012	TT	≥20	363	Ref	445	Ref	331	Ref
	GT	≥20	372	1.14 (0.40–3.23)	510	1.24 (0.53–2.93)	405	1.12 (0.42–2.94)
	GG	≥20	121	2.67 (0.86–8.29)	151	1.84 (0.60–5.61)	115	2.56 (0.83–7.85)
	TT	<20	661	1.28 (0.52–3.15)	361	1.08 (0.41–2.84)	168	0.84 (0.21–3.27)
	GT	<20	725	1.41 (0.59–3.41)	370	1.26 (0.51–3.12)	180	1.32 (0.41–4.21)
	GG	<20	201	1.79 (0.58–5.49)	114	2.80 (0.97–8.10)	54	2.26 (0.45–11.45)
	*P* _interaction_			0.6139		0.4639		0.9865
** *CYP3A4* **								
rs2242480	CC	≥20	490	Ref	644	Ref	504	Ref
	CT	≥20	314	0.89 (0.34–2.31)	404	1.16 (0.51–2.64)	293	1.08 (0.44–2.68)
	TT	≥20	54	1.74 (0.37–8.17)	59	1.57 (0.35–7.15)	52	2.47 (0.66–9.19)
	CC	<20	899	1.04 (0.51–2.13)	480	1.53 (0.75–3.13)	247	1.25 (0.48–3.22)
	CT	<20	604	1.16 (0.54–2.47)	318	0.53 (0.17–1.61)	142	0.51 (0.11–2.33)
	TT	<20	87	1.52 (0.41–5.62)	51	3.13 (0.85–11.47)	17	3.30 (0.39–28.05)
	*P* _interaction_			0.7911		0.696		0.5117
** *GC* **								
rs1155563	TT	≥20	331	Ref	440	Ref	326	Ref
	TC	≥20	401	0.81 (0.31–2.11)	498	1.04 (0.44–2.44)	392	0.45 (0.17–1.15)
	CC	≥20	127	0.81 (0.21–3.09)	167	1.31 (0.44–3.95)	133	0.87 (0.27–2.80)
	TT	<20	531	1.26 (0.55–2.88)	272	1.70 (0.70–4.15)	123	0.21 (0.03–1.65)
	TC	<20	766	0.78 (0.34–1.80)	403	1.02 (0.40–2.55)	199	0.92 (0.33–2.52)
	CC	<20	290	0.89 (0.33–2.42)	172	0.87 (0.26–2.89)	83	0.94 (0.25–3.54)
	*P* _interaction_			0.9456		0.5655		0.2934
rs16846876	AA	≥20	437	Ref	557	Ref	419	Ref
	AT	≥20	366	1.22 (0.48–3.07)	450	1.18 (0.52–2.70)	353	0.34 (0.12–0.99)
	TT	≥20	58	1.31 (0.27–6.38)	102	1.46 (0.43–4.97)	81	1.83 (0.62–5.37)
	AA	<20	714	1.65 (0.75–3.60)	379	1.71 (0.77–3.79)	168	0.51 (0.14–1.84)
	AT	<20	672	0.87 (0.37–2.07)	362	1.00 (0.40–2.51)	182	1.00 (0.38–2.69)
	TT	<20	206	1.12 (0.37–3.36)	108	0.72 (0.15–3.38)	55	0.48 (0.06–3.85)
	*P* _interaction_			0.4024		0.3482		0.8437
rs17467825	AA	≥20	447	Ref	578	Ref	430	Ref
	GA	≥20	352	0.79 (0.31–1.99)	437	1.06 (0.48–2.34)	337	0.41 (0.15–1.07)
	GG	≥20	62	0.49 (0.06–3.95)	94	0.59 (0.12–3.00)	88	0.56 (0.13–2.33)
	AA	<20	692	1.15 (0.55–2.38)	347	1.55 (0.71–3.40)	152	0.36 (0.08–1.61)
	GA	<20	691	0.91 (0.43–1.93)	383	0.84 (0.35–2.04)	191	0.85 (0.32–2.25)
	GG	<20	213	0.65 (0.21–1.98)	121	0.81 (0.22–2.96)	62	0.76 (0.17–3.48)
	*P* _interaction_			0.8118		0.8253		0.2029
rs2282679	TT	≥20	449	Ref	578	Ref	429	Ref
	GT	≥20	349	0.81 (0.32–2.04)	434	1.27 (0.57–2.80)	336	0.42 (0.16–1.10)
	GG	≥20	63	0.47 (0.06–3.80)	98	0.62 (0.12–3.11)	89	0.56 (0.13–2.34)
	TT	<20	688	1.10 (0.53–2.29)	345	1.67 (0.75–3.71)	153	0.36 (0.08–1.60)
	GT	<20	689	0.97 (0.46–2.04)	381	0.92 (0.37–2.25)	188	0.87 (0.33–2.30)
	GG	<20	217	0.65 (0.21–1.97)	123	0.88 (0.24–3.22)	63	0.76 (0.17–3.48)
	*P* _interaction_			0.7444		0.6979		0.2075
rs2298849	AA	≥20	332	Ref	430	Ref	346	Ref
	GA	≥20	411	3.12 (1.02–9.59) *	511	0.92 (0.41–2.05)	386	0.93 (0.38–2.27)
	GG	≥20	120	1.36 (0.24–7.60)	170	0.43 (0.09–1.95)	124	1.23 (0.37–4.10)
	AA	<20	692	2.29 (0.77–6.85)	378	0.72 (0.29–1.82)	173	1.01 (0.33–3.11)
	GA	<20	700	2.33 (0.78–6.95)	362	1.28 (0.57–2.87)	178	1.25 (0.41–3.82)
	GG	<20	204	1.94 (0.51–7.40)	111	1.01 (0.28–3.70)	54	—
	*P* _interaction_			0.5918		0.1481		0.4554
rs2298850	GG	≥20	441	Ref	564	Ref	416	Ref
	CG	≥20	354	0.77 (0.31–1.95)	437	1.34 (0.60–3.01)	341	0.41 (0.15–1.10)
	CC	≥20	63	0.46 (0.06–3.71)	99	0.66 (0.13–3.33)	89	0.56 (0.13–2.37)
	GG	<20	672	1.04 (0.50–2.20)	339	1.83 (0.81–4.12)	150	0.38 (0.09–1.70)
	CG	<20	694	0.90 (0.42–1.91)	385	0.85 (0.33–2.20)	191	0.75 (0.26–2.12)
	CC	<20	215	0.64 (0.21–1.96)	122	0.94 (0.25–3.50)	63	0.79 (0.17–3.65)
	*P* _interaction_			0.759		0.5603		0.2452
rs3755967	CC	≥20	448	Ref	575	Ref	427	Ref
	CT	≥20	351	0.80 (0.32–2.01)	438	1.25 (0.57–2.76)	340	0.41 (0.16–1.08)
	TT	≥20	63	0.47 (0.06–3.78)	98	0.61 (0.12–3.10)	89	0.56 (0.13–2.32)
	CC	<20	685	1.09 (0.52–2.29)	345	1.66 (0.75–3.69)	153	0.36 (0.08–1.58)
	CT	<20	695	0.96 (0.46–2.02)	383	0.91 (0.37–2.24)	189	0.86 (0.33–2.27)
	TT	<20	217	0.64 (0.21–1.96)	123	0.87 (0.24–3.20)	63	0.75 (0.16–3.45)
	*P* _interaction_			0.7354		0.7072		0.2031
rs4588	GG	≥20	448	Ref	571	Ref	422	Ref
	GT	≥20	348	0.81 (0.32–2.05)	439	1.23 (0.56–2.72)	339	0.40 (0.15–1.07)
	TT	≥20	64	0.45 (0.06–3.57)	98	0.60 (0.12–3.05)	90	0.54 (0.13–2.24)
	GG	<20	676	1.11 (0.53–2.32)	341	1.66 (0.75–3.70)	149	0.37 (0.08–1.65)
	GT	<20	696	0.91 (0.43–1.93)	382	0.80 (0.31–2.05)	192	0.70 (0.25–1.98)
	TT	<20	215	0.65 (0.21–1.99)	124	0.84 (0.23–3.11)	63	0.75 (0.16–3.46)
	*P* _interaction_			0.8164		0.6081		0.2578
rs7041	AA	≥20	423	Ref	573	Ref	454	Ref
	CA	≥20	358	2.46 (0.96–6.32)	453	0.92 (0.42–2.04)	337	0.65 (0.27–1.58)
	CC	≥20	82	0.97 (0.12–8.11)	84	0.55 (0.07–4.25)	63	0.50 (0.06–3.91)
	AA	<20	892	2.03 (0.87–4.75)	478	1.15 (0.55–2.40)	226	0.80 (0.30–2.13)
	CA	<20	612	1.21 (0.46–3.14)	316	1.04 (0.43–2.50)	158	0.61 (0.17–2.18)
	CC	<20	88	2.18 (0.54–8.77)	56	—	21	—
	*P* _interaction_			0.2275		0.8097		0.8926
** *LRP2* **								
rs10210408	CC	≥20	289	Ref	354	Ref	287	Ref
	TC	≥20	418	1.75 (0.60–5.10)	558	1.45 (0.58–3.62)	415	0.69 (0.29–1.62)
	TT	≥20	155	1.56 (0.40–6.00)	199	1.25 (0.39–4.03)	154	0.30 (0.07–1.42)
	CC	<20	526	1.83 (0.66–5.12)	294	1.10 (0.36–3.33)	127	0.41 (0.09–1.94)
	TC	<20	782	1.64 (0.61–4.40)	399	1.96 (0.79–4.89)	214	0.86 (0.32–2.31)
	TT	<20	290	1.16 (0.34–3.91)	158	0.64 (0.13–3.15)	64	0.44 (0.06–3.55)
	*P* _interaction_			0.497		0.9331		0.3864
rs2228171	TT	≥20	291	Ref	374	Ref	292	Ref
	CT	≥20	92	0.97 (0.19–4.86)	89	2.33 (0.65–8.38)	63	0.98 (0.21–4.70)
	CC	≥20	84	1.61 (0.39–6.59)	104	0.91 (0.19–4.49)	74	0.37 (0.05–3.00)
	TT	<20	548	1.21 (0.48–3.07)	294	1.06 (0.36–3.16)	140	0.60 (0.16–2.29)
	CT	<20	286	1.68 (0.62–4.56)	96	2.12 (0.59–7.56)	60	—
	CC	<20	194	1.26 (0.38–4.14)	111	2.37 (0.67–8.41)	59	1.22 (0.25–5.97)
	*P* _interaction_			0.9483		0.4054		0.5358
rs2389557	AA/GA	≥20	113	Ref	102	Ref	87	Ref
	GG	≥20	37	2.15 (0.18–25.61)	35	1.66 (0.22–12.36)	26	2.11 (0.21–20.73)
	AA/GA	<20	433	2.50 (0.55–11.48)	156	1.04 (0.20–5.46)	100	0.55 (0.07–4.57)
	GG	<20	125	3.57 (0.70–18.10)	33	7.09 (1.21–41.47)	18	—
	*P* _interaction_			0.7002		0.8493		0.2569
rs4667591	TT/GT	≥20	120	Ref	116	Ref	94	Ref
	GG	≥20	31	7.98 (0.64–98.90)	21	6.84 (0.76–61.89)	19	0.90 (0.04–21.95)
	TT/GT	<20	458	4.84 (0.63–37.13)	146	1.74 (0.35–8.58)	93	0.15 (0.01–2.10)
	GG	<20	100	7.64 (0.88–66.52)	44	7.44 (1.11–49.79) *	26	1.60 (0.13–19.78)
	*P* _interaction_			0.8345		0.7612		0.3993
** *VDR* **								
rs10783219	AA	≥20	309	Ref	431	Ref	336	Ref
	TA	≥20	412	1.15 (0.44–3.05)	523	0.80 (0.36–1.77)	397	1.66 (0.67–4.12)
	TT	≥20	140	0.96 (0.24–3.87)	156	0.41 (0.09–1.90)	120	0.77 (0.16–3.86)
	AA	<20	602	1.28 (0.52–3.15)	297	0.78 (0.30–2.02)	145	1.44 (0.41–5.02)
	TA	<20	739	1.15 (0.47–2.78)	419	1.00 (0.45–2.21)	194	0.71 (0.18–2.78)
	TT	<20	253	0.90 (0.29–2.80)	135	1.04 (0.33–3.31)	65	2.65 (0.66–10.67)
	*P* _interaction_			0.8452		0.1733		0.945
rs2238136	CC	≥20	111	Ref	99	Ref	83	Ref
	TC/TT	≥20	40	4.94 (0.42–58.60)	38	6.78 (0.84–54.44)	30	0.77 (0.06–9.56)
	CC	<20	363	5.26 (0.68–40.67)	126	4.10 (0.66–25.32)	77	0.44 (0.06–3.28)
	TC/TT	<20	196	3.82 (0.45–32.40)	65	2.57 (0.29–22.94)	42	—
	*P* _interaction_			0.4588		0.0586		0.9648

Abbreviations: VitD, vitamin D; HDP, hypertensive disorders in pregnancy. Adjusted for pre-pregnancy BMI, maternal age, educational level, parity, basal blood pressure. * *p* < 0.05.

## Data Availability

The data presented in this study are available on request from the corresponding author. The data are not publicly available because they contain information that could compromise the privacy of research participants.

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
