# Peer review of "The Association of Vitamin D and Its Pathway Genes’ Polymorphisms with Hypertensive Disorders of Pregnancy: A Prospective Cohort Study"

_nutrients, 2022, doi:10.3390/nu14112355_

Round 1
Reviewer 1 Report
First of all congratulations for the work done.
1.The introduction is very contextualized.
2. Could the authors please what tool did you use to assess biases?
3. It has a great clinical impact.
Thank you very much.
Author Response
Point 1: Could the authors please what tool did you use to assess biases?
Response 1: As we stated in the limitation part, line 346-348, some subjects were lack data of 25(OH)D at T2 and T3, selection bias might exist. However, subgroup analysis conducted among pregnant women with VitD detected at T1 and T2 showed that the results were consistent with the results in the whole study population. In addition, as it was a prospective cohort, recall bias could be avoided. Furthermore, plasma 25(OH)D2 and 25(OH)D3 concentrations were measured by liquid chromatography–tandem mass spectrometry (API 3200MD (Applied Bio-systems/MDS Sciex, USA)), which is the gold standard of measuring VitD. Questionnaire investigation was performed by well-trained interviewers. So, the measurement bias could be minimal. For confounding bias, multivariate regression model was used for adjusting the potential confounding effects.
Reviewer 2 Report
Si et al. Presented a manuscript on the association between vitamin D and hypertensive disorders in pregnancy. There are few comments this reviewer has:
- Authors haven't described the blood pressure monitoring in methods. This has be to be more thoroughly described as it is one of the primary outcomes from the study.
- Authors state in line 226 - after adjusting confounders. What confounders are you discussing there?
- One of the confounders is the weight of the subjects. As shown in Table 1 HDP patients weighed significantly higher than non-HDP subjects. How authors describe this confounder?
- PE is not a generic abbreviation for pre-eclampsia. PE is mostly used for pulmonary embolism.
- Line 271 - Authors state that "polymorphism in CYP24A1, GC and LRP2 were associated with blood pressure and HDP". What data were taken as a rationale for this statement?
- Authors use a case control study as an example, however do not provie citation in line 312.
Author Response
Point 1: Authors haven't described the blood pressure monitoring in methods. This has be to be more thoroughly described as it is one of the primary outcomes from the study.
Response 1: The blood pressure monitoring was a routine part of prenatal health care. Pregnant women received blood pressure measurement at the first check-up as well as the following check-ups before delivery, and the data of blood pressure was recorded in the electronic medical recorder system, please see section 2.4 Data extraction, line 117-128. The outcome of this study was based on the data extracted from electronic medical recorder system. In addition, the method of blood pressure measurement was described in section 2.6 HDP Definition, line 142-150.
Point 2: Authors state in line 226 - after adjusting confounders. What confounders are you discussing there?
Response 2: Thanks for your comments. The adjusting confounders were pre-pregnancy BMI, maternal age, gestational weight gain, gestational week, educational level, parity, basal blood pressure and the seasons of blood pressure measurement, which has been described in section 2.8 Statistical Analysis, line 195-198.
Point 3: One of the confounders is the weight of the subjects. As shown in Table 1 HDP patients weighed significantly higher than non-HDP subjects. How authors describe this confounder?
Response 3: It has been revised in the manuscript. “Compared with non-HDP participants, HDP women had higher pre-pregnancy BMI (21.16±2.91 kg/m2 vs. 23.62 ± 4.05 kg/m2, P < .0001)”. In addition, we also adjusted pre-pregnancy BMI and gestational weight gain in multivariate regression models, please see section 2.8 Statistical Analysis, line 195-198.
Point 4: PE is not a generic abbreviation for pre-eclampsia. PE is mostly used for pulmonary embolism.
Response 4: Thanks for your suggestion. We have revised it through the whole manuscript.
Point 5: Line 271 - Authors state that "polymorphism in CYP24A1, GC and LRP2 were associated with blood pressure and HDP". What data were taken as a rationale for this statement?
Response 5: Sorry for the unclear statement. It has been revised in the manuscript: “Polymorphism in CYP24A1, GC and LRP2 were associated with blood pressure and polymorphism in CYP24A1 and GC were associated with increased risk of HDP.”
Point 6: Authors use a case control study as an example, however do not provie citation in line 312.
Response 6: Thanks for your suggestion. It is the same citation of 39. It has been added in the manuscript.
Reviewer 3 Report
Regarding the article “The Association of Vitamin D and Its Pathway Genes’ Polymorphisms with Hypertensive Disorders of Pregnancy: A Prospective Cohort Study”, the authors did not minimally comply with the main rule of the journal, which is to comply with the rules. The citations are incorrect and the tables are completely unformatted. The authors did not take care to adjust the article to the journal's rules. The manuscript is confused and has a damaged aesthetic.
The article is not registered in Clinical Trials Registration
https://www.mdpi.com/journal/nutrients/instructions
The authors say “This prospective cohort study was based on the data of Zhoushan Pregnant Women 86 Cohort (ZPWC) from August 2011 to May 2018, which is an ongoing prospective cohort 87 conducted in Zhoushan Maternal and Child Health Care Hospital, Zhejiang.” – however this is not a correct study design as this study has already been developed. Recruitment of participants started in 2011 and ended in 2018 ? Why was there this window between 2019 to 2022?
English needs major adjustments.
Author Response
Point 1: Regarding the article “The Association of Vitamin D and Its Pathway Genes’ Polymorphisms with Hypertensive Disorders of Pregnancy: A Prospective Cohort Study”, the authors did not minimally comply with the main rule of the journal, which is to comply with the rules. The citations are incorrect and the tables are completely unformatted. The authors did not take care to adjust the article to the journal's rules. The manuscript is confused and has a damaged aesthetic.
Response 1: Sorry for this. We have revised the manuscript and formatted the tables according to the main rules of the journal.
Point 2: The article is not registered in Clinical Trials Registration. https://www.mdpi.com/journal/nutrients/instructions
Response 2: Yes, we don’t register it in Clinical Trials Registration, due thatit is an observational cohort study. According to the “Instructions for Authors”, purely observational studies do not require registration. So, there is no need to register in Clinical Trials Registration.
Point 3: The authors say “This prospective cohort study was based on the data of Zhoushan Pregnant Women 86 Cohort (ZPWC) from August 2011 to May 2018, which is an ongoing prospective cohort 87 conducted in Zhoushan Maternal and Child Health Care Hospital, Zhejiang.” – however this is not a correct study design as this study has already been developed. Recruitment of participants started in 2011 and ended in 2018 ? Why was there this window between 2019 to 2022?
Response 3: Sorry for the mistake. The Ethical Approval Code provided in the manuscript was not for this study. We have revised it:“The study was conducted according to the guidelines of the Declaration of Helsinki, and approved by the Institutional Review Board of Zhoushan Maternal and Child Health Care Hospital on 9 January 2011 (Ethical Approval Code: 2011-05).”
Point 4: English needs major adjustments.
Response 4: Thanks for the suggestion. We have carefully checked and revised English language and style in the manuscript.
Round 2
Reviewer 3 Report
Received. The article can be accepted for publication.